**eLife** | RESEARCH ARTICLE

# Mechanical forces pattern endocardial Notch activation via mTORC2-PKC pathway

**Yunfei Mu**[1,2,3,4†], **Shijia Hu**[1,2,3,4†], **Xiangyang Liu**[2,3,4], **Xin Tang**[2,3,4], **Jiayi Lin**[2,3,4], **Hongjun Shi**[2,3,4]*

[1]Fudan University, Shanghai, China; [2]Key Laboratory of Growth Regulation and Translational Research of Zhejiang Province, School of Life Sciences, Westlake University, Hangzhou, China; [3]Westlake Laboratory of Life Sciences and Biomedicine, Hangzhou, China; [4]Institute of Basic Medical Sciences, Westlake Institute for Advanced Study, Hangzhou, China

## eLife Assessment

Notch1 is expressed uniformly throughout the mouse endocardium during the initial stages of heart valve formation, yet it remains unclear how Notch signaling is activated specifically in the AVC region to induce valve formation. To answer this question, the authors used a combination of in vivo and ex vivo experiments in mice to demonstrate ligand-independent activation of Notch1 by circulation induced-mechanical stress and provide evidence for stimulation of a novel mechanotransduction pathway involving post-translational modification of mTORC2 and Protein Kinase C (PKC) upstream of Notch1. These findings represent an **important** advance in our understanding of valve formation and the conclusions are supported by **convincing** data.

*For correspondence:
shihongjun@westlake.edu.cn

†These authors contributed equally to this work

Competing interest: The authors declare that no competing interests exist.

**Abstract** Notch signaling has been identified as a key regulatory pathway in patterning the endocardium through activation of endothelial-to-mesenchymal transition (EMT) in the atrioventricular canal (AVC) and proximal outflow tract (OFT) region. However, the precise mechanism underlying Notch activation remains elusive. By transiently blocking the heartbeat of E9.5 mouse embryos, we found that Notch activation in the arterial endothelium was dependent on its ligand Dll4, whereas the reduced expression of Dll4 in the endocardium led to a ligand-depleted field, enabling Notch to be specifically activated in AVC and OFT by regional increased shear stress. The strong shear stress altered the membrane lipid microdomain structure of endocardial cells, which activated mTORC2 and PKC and promoted Notch1 cleavage even in the absence of strong ligand stimulation. These findings highlight the role of mechanical forces as a primary cue for endocardial patterning and provide insights into the mechanisms underlying congenital heart diseases of endocardial origin.

## Introduction

The formation of the mammalian heart begins with a simple linear heart tube which undergoes complex morphogenic processes to ultimately reach the adult four-chamber configuration. In mice, the pattern of the adult heart becomes apparent at embryonic day (E) 9.5. At this stage, increased cell proliferation and sarcomeric development at the outer curvature of the primary heart tube lead to the formation of working chamber myocardium, while less proliferative and less differentiated myocardium at the inner curvature characterizes the OFT and the AVC (*Moorman and Christoffels, 2003*; *Miquerol and Kelly, 2013*). Between E9.5 and E10.5, endocardial cells in the OFT and AVC regions

undergo extensive EMT, forming endocardial cushions. Further growth and remodeling of these endocardial cushions contribute to the septation of the OFT and cardiac chambers and give rise to cardiac valves (*Leckband et al., 2011*). Malformation of the endocardial cushions underlies the mechanism of the majority of human congenital heart defects (CHD).

Multiple interactive signal transduction pathways are involved in patterning valve versus chamber myocardium and regulating EMT. At E9.5, expression of BMP2 in the AVC and OFT myocardium (*Ma et al., 2005*; *Rivera-Feliciano and Tabin, 2006*) directly activates TBX2 expression, which, in turn, activates TGFβ2 expression (*Shirai et al., 2009*). Both BMP2 and TGF-β2 enhance Snail1 expression and stimulate EMT (*Ma et al., 2005*; *Niessen et al., 2008*). Notch signaling is also required for EMT. At E9.5, NICD (Notch1 intracellular domain) expression is the highest in endocardial cells of the AVC and OFT region, while in the ventricle, NICD is more restricted to the endocardium at the base of developing trabeculae (*Del Monte et al., 2007*; *Grego-Bessa et al., 2007*). Notch, through the transcriptional factor CSL, directly activates Snail2 expression, leading to the repression of VE-cadherin expression (*Niessen et al., 2008*; *Timmerman et al., 2004*). Genetic abrogation of Notch signaling blocks EMT in the AVC endocardium, whereas ectopic endocardial NICD expression leads to partial EMT of ventricular endocardial cells outside AVC (*Timmerman et al., 2004*; *Luna-Zurita et al., 2010*).

Despite the well-characterized role of Notch in endocardial patterning and EMT, little is known about the mechanism that specifically activates Notch in the areas of endocardium that undergo EMT. Among various Notch receptors and ligands, Notch1, Dll4, and Jag1 are expressed in the endocardium at the onset of EMT (*Timmerman et al., 2004*; *Krebs et al., 2000*). Notch1 transcript appears to be uniformly expressed throughout the endocardium at E9.5, while Dll4 is most concentrated in the ventricular endocardium at the base of the trabeculae (*Grego-Bessa et al., 2007*). Jag1 is highly expressed throughout the myocardium, with only a few individual AVC endocardial cells positive for Jag1 (*MacGrogan et al., 2016*; *Luxán et al., 2016*). Conditional deletion of Dll4 in all endothelial cells results in the loss of classic EMT markers and the complete absence of AVC cushions at E9.5 (*MacGrogan et al., 2016*). However, it remains unclear whether this impact on EMT is specifically due to the loss of Dll4-Notch signaling at cushion endocardium or a consequence of global circulatory failure resulting from vascular deformation in Dll4-deficient embryos. Therefore, the currently known expression pattern of Notch receptors and ligands does not fully explain the specific pattern of Notch activation in the endocardium.

Studies in zebrafish embryos have revealed the essential role of various mechanosensitive ion channels in modulating Notch1b receptor expression in the AVC endocardium in response to increased shear stress in this area (*Duchemin et al., 2019*; *Gálvez-Santisteban et al., 2019*; *Heckel et al., 2015*). However, as mentioned earlier, in the developing mouse endocardium at the onset of EMT, Notch1 is uniformly expressed throughout the endocardium despite restricted NICD in AVC and OFT endocardium. Thus, the equivalent mechanosensitive pathways controlling region-specific patterns of Notch activation, and hence EMT, in the mammalian endocardium are still unclear. To address this question, we transiently blocked the heartbeat of mouse embryos at E9.5 in vivo using the class III antiarrhythmic drug dofetilide, which selectively blocks the rapid component of the delayed rectifier $K^+$ current ($I_{Kr}$) in cardiomyocytes (*Mounsey and DiMarco, 2000*). We found that Notch activation in the vascular endothelium is dependent on Dll4, whereas in the endocardium, the overall reduction of Dll4 expression creates a ligand-depleted field that allows the establishment of a specific Notch activation pattern in the AVC and OFT regions in response to increased shear stress. Strong shear stress in these valve-forming areas alters the membrane lipid microdomain structure of the endocardial cells, which then activates mTORC2 and PKC, promoting Notch1 cleavage even in the absence of strong ligand stimulation. The results uncovered a new mechanism whereby mechanical force serves as a primary cue for endocardial patterning in mammalian embryonic heart.

## Results

### Notch is strongly activated in the AVC and proximal OFT endocardium in spite of weak ligand expression at the onset of EMT

Although Notch1, Dll4, and Jag1 have been reported as the principal receptor and ligands expressed in the endocardium (*Krebs et al., 2000*; *MacGrogan et al., 2016*; *D'Amato et al., 2016*), the relationship between the Notch activity and its ligand expression in various parts of the E8.5–9.5 endocardium

has not been examined in sufficient details. To investigate whether regional-specific Notch activation is due to differential receptor or ligand expression, we performed whole-mount in situ hybridization and immunofluorescence staining of Notch1, NICD, Dll4, and Jag1. Both *Dll4* transcripts and NICD were uniformly positive throughout all endothelial lining of the dorsal aorta and heart at E8.5 (*Figure 1—figure supplement 1A*, *Video 1* and *Video 2*). However, at E9.5 when endocardial EMT starts, distinct patterns of NICD and Dll4 expression appeared (*Figure 1B* and *Videos 3 and 4*). Based on the expression pattern of NICD and Dll4, overall, the endocardium at E9.5 can be viewed as three distinct types as summarized in *Figure 1A*. Type I is Dll4-high and NICD-high, including the endocardium lining the distal OFT, the base of the trabeculae, the dorsal wall of the left atrium, and the right atrium. These endocardial cells normally do not undergo EMT. Type II is Dll4-low and NICD-high, including the AVC and proximal OFT endocardium. Endocardial cells in these regions undergo EMT to form endocardial cushion cells. Type III is Dll4-low and NICD-low, including the endocardium flanking the AVC and on the top of ventricular trabeculae (*Figure 1B* and *Figure 1—figure supplement 1C*). Jag1 was highly expressed in the myocardium but expressed at a very low level in a small subset of the OFT and AVC endocardial cells (*Figure 1—figure supplement 1A*), consistent with the previous report (*MacGrogan et al., 2016*). Vascular endothelium continued to express high levels of Dll4 and NICD uniformly, similar to type I endocardial cells. Dll4 protein staining pattern overlapped with the *Dll4* transcript and also agreed with VEGF receptor-2 (Flk1/KDR) protein expression throughout the cardiovascular endothelium at both E8.5 and E9.5 (*Figure 1—figure supplement 1B*), consistent with the regulatory role of VEGF signaling in Dll4 expression (*Wythe et al., 2013*). Thus, the Dll4 and NICD expression patterns were disconnected in the type II endocardium at E9.5, suggesting the existence of additional mechanisms for Notch activation.

One previous study showed that endocardial EMT was dependent on Dll4 (*MacGrogan et al., 2016*). To further dissect the role of Dll4 in regional Notch activation, we conditionally deleted Dll4 in all endothelial cells using the *Tek-cre* line. At E9.5, *Dll4* conditional null AVC cushions were poorly cellularized, consistent with the previous finding. Furthermore, in most areas normally expressing high Dll4, i.e., vascular endothelium and aortic sac, NICD was completely abolished, whereas in the AVC, proximal OFT, and the base of the trabeculae, NICD was reduced but not completely absent (*Figure 1C*). As conditional deletion of Dll4 resulted in embryos with severely disorganized vasculature (*Figure 1C*), we cannot rule out the possibility that the reduced NICD in the AVC might not be simply due to Dll4 deletion, but rather affected by circulatory failure. These findings indicate that vascular endothelium is dependent on Dll4 for Notch activation, whereas in the endocardium, additional factors are required to pattern Notch activation in the proximal OFT and AVC.

## Notch activation in the cushion endocardium is dependent on blood flow

Work in the past on zebrafish embryos showed that cardiac contraction activates endocardial Notch signaling *Samsa et al., 2015*. To test if Notch activation is regulated by blood flow in the developing mouse heart, pregnant mice were gavaged with a single dose of dofetilide at 2 mg/kg at E9.5. The embryos were then either harvested immediately after the treatment or allowed to develop until E18.5 when cardiac morphology was assessed (*Figure 2—figure supplement 1A*). Dofetilide caused an immediate stop of blood flow in the embryos, as observed by doppler ultrasound from 1 hr through to 3 hr, which completely recovered at 5 hr post-treatment (*Figure 2A* and *Videos 5 and 6*). Three hours of cessation of blood flow caused a complete loss of NICD without affecting the total Notch1 receptor protein in the proximal OFT and AVC endocardium. The loss of NICD was completely recovered at 5 hr post-treatment, in line with the dynamics of the heart rate changes (*Figure 2B*), while no significant changes in the expressions of Dll4 and Jag1 were noted in AVC endocardium after dofetilide treatment (*Figure 2B*). Pro-EMT markers phospho-Smad1/5, Sox9, and Twist1 were downregulated in the AVC endocardium after cardiac arrest (*Figure 2B*). Interestingly, NICD in the dorsal aorta was resistant to the cessation of flow (*Figure 2—figure supplement 1B*). Consistent with the inhibition of EMT, transient cessation of blood flow resulted in hypoplastic AVC endocardial cushions 5 hr after treatment (*Figure 2—figure supplement 1C*) and more pronounced cushion hypoplasia 1 d after treatment (*Figure 2C*), and various heart defects (*Figure 2D* and *Figure 2—figure supplement 1D*) in 40% of embryos: ventricular septal defects (VSD), bicuspid semilunar valve, atrioventricular valve defects, and conotruncal defects, consistent with malformation of endocardial cushions.

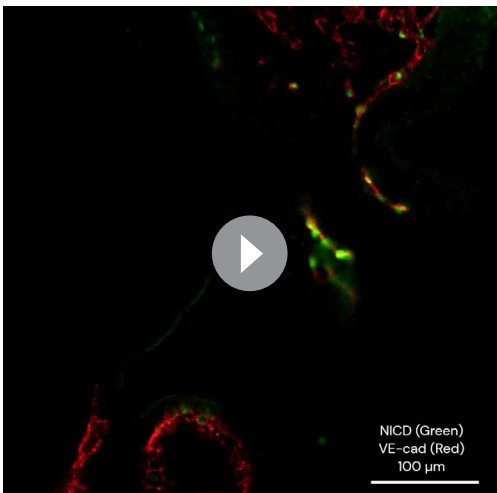

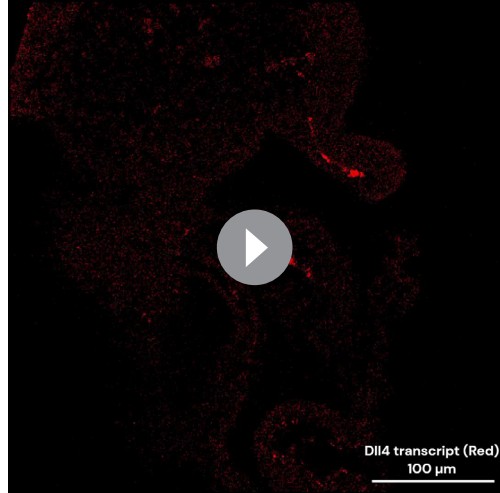

**Video 1.** E8.5 NICD whole mount staining. NICD (green) whole-mount immunofluorescence staining of E8.5 mouse heart.

https://elifesciences.org/articles/97268/figures#video1

**Video 2.** E8.5 Dll4 in situ hybridization. Whole-mount *Dll4* (red) in situ hybridization staining of E8.5 mouse heart.

https://elifesciences.org/articles/97268/figures#video2

To rule out the possible direct effect of dofetilide on endocardial cells independent of flow, we conditionally deleted *Tnnt2* in the embryonic myocardium, which caused bradycardia in the E9.5 embryos and lethality at E10.5 (*Video 7*). Similar to the effect of dofetilide, reducing blood flow rate by genetic means also caused a significant reduction of NICD in the cushion endocardium at E9.5 (*Figure 2—figure supplement 2A*). Furthermore, the block of heartbeat by ex vivo blebbistatin treatment, an inhibitor for non-muscle myosin II ATPase, also prevented Notch cleavage in the cushion endocardium at E9.5 (*Figure 2—figure supplement 2B*). To rule out the possible effect of hypoxia on Notch activation, we depleted embryonic erythrocytes by crossing *Epor^{P2A-icre}* with *ROSA-DTA* mouse line, which resulted in widespread hypoxia without interfering with embryonic heartbeat, yet NICD was normal in the AVC endocardium (*Figure 2—figure supplement 2C*). We also cultured E9.5 embryos in a medium saturated with 95% $O_2$ in the presence of dofetilide for 3 hr. High levels of $O_2$ eliminated HIF-1α nuclear staining induced by cardiac arrest, while NICD was still absent in the AVC endocardium (*Figure 2—figure supplement 2D*). Thus, all subsequent ex vivo embryo culture experiments were performed in 95% $O_2$ to minimize the effect of hypoxia on Notch activation. Collectively, these results indicate that Notch activation and EMT in the cushion endocardium are dependent on the mechanical stimulation from blood flow.

## Flow-responsive mTORC2-PKCε activity is required for Notch activation in the cushion endocardium

Given the fast response of NICD to flow cessation, we suspect a change in post-translational modification, such as a phosphorylation event, that might mediate the effect. PKC, AKT, and ERK have all been reported to have their phosphorylation status altered in response to fluid shear stress in cultured endothelial cells (*Li et al., 2005*). In addition, in vitro, shear stress-induced Notch activation can be blocked by inhibitors of PKC, AKT, and ERK (*Masumura et al., 2009*). Therefore, we examined the phosphorylation status of these three signaling molecules in the endocardium before and after dofetilide treatment. Both pPKC (βII^{Ser660}) and pAKT^{Ser473} were restricted to the cushion endocardium and the base of the trabeculae in control embryos and were almost completely lost as quickly as 1 hr after treatment and then both recovered at 5 hr after treatment. Both pPKC and pAKT were not detectable in the dorsal aorta endothelium (*Figure 3A*, *Figure 3—figure supplement 1A*). Their response rate in the cushion endocardium was faster than NICD, whose maximum inhibition occurred at 3 hr post-treatment (*Figure 2B*). pERK was not detectable in the cushion endocardium (*Figure 3—figure supplement 1B*). Inhibition of AKT phosphorylation in cultured E9.5 embryos by wortmannin did not inhibit Notch activation (*Figure 3—figure supplement 1C*), whereas inhibition of PKC activity by staurosporine treatment blocked Notch activation (*Figure 3B*).

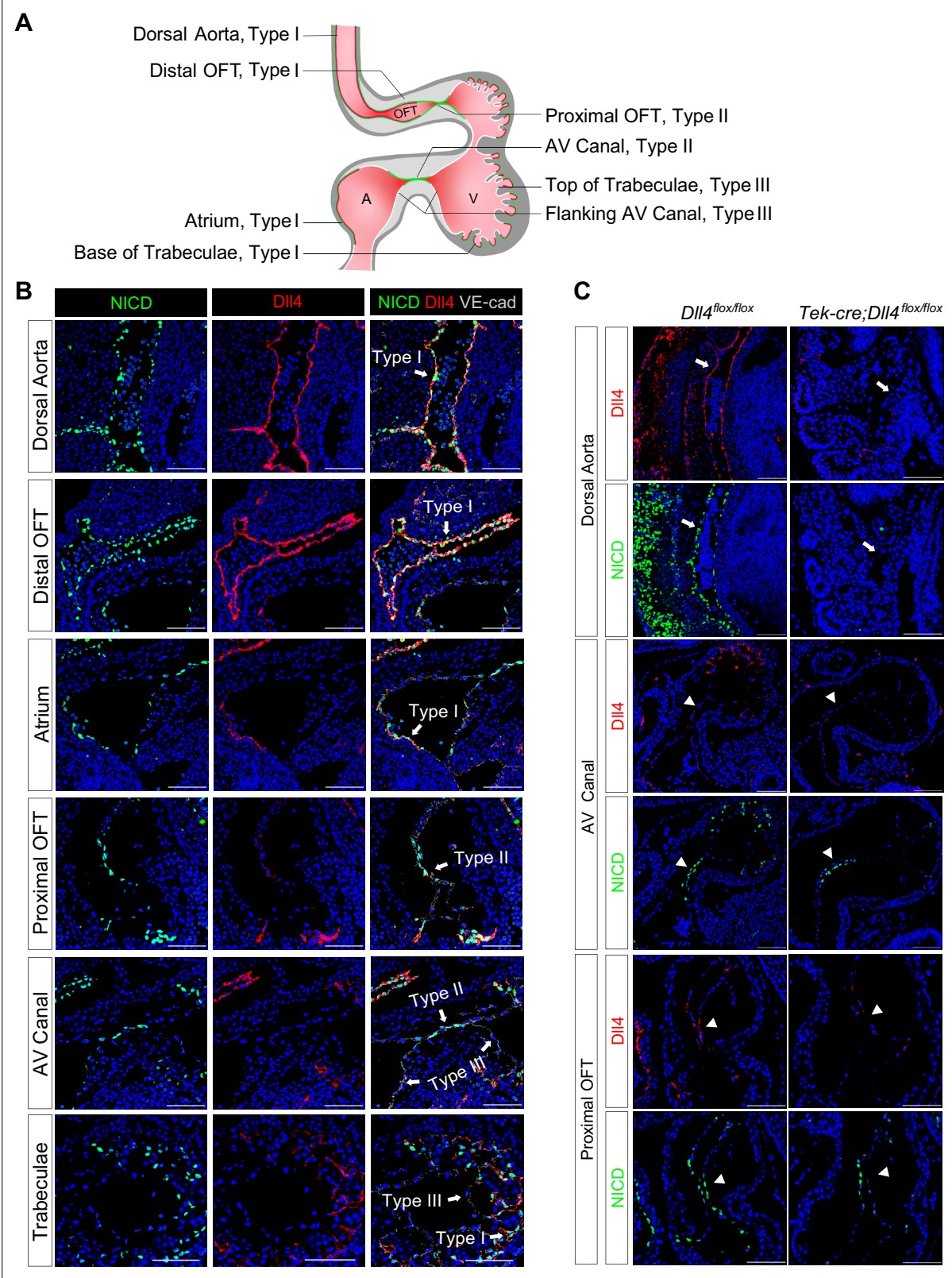

**Figure 1.** Endocardium and vascular endothelium of dorsal aorta showed different patterns of Notch1 activation and Dll4 ligand expression. (**A**) Schematic representation of three types of endothelial cells and their corresponding localizations. (**B**) Type I cells are NICD-high and Dll4-high. Type II cells are NICD-high and Dll4-low. Type III cells are NICD-low and Dll4-low. Scale bars, 100 μm. (n = 3 embryos) (**C**) Representative images showing Dll4

*Figure 1 continued on next page*

*Figure 1 continued*

protein and NICD in wild-type and endothelial-specific *Dll4*-deleted (*Tek-cre; Dll4flox/flox*) E9.5 mouse hearts. Scale bars, 100 µm. (wild-type: n = 3 embryos; *Tek-cre; Dll4flox/flox*: n = 3 embryos).

The online version of this article includes the following figure supplement(s) for figure 1:

**Figure supplement 1.** Notch is strongly activated in the atrioventricular canal (AVC) and proximal outflow tract (OFT) endocardium despite weak ligand expression at the onset of endothelial-to-mesenchymal transition (EMT).

To further confirm the role of PKC in regulating Notch, we individually knocked out *Prkce* and *Prkch* (*Figure 3—figure supplement 2C*). Among all PKC family members, these two isozymes are most highly expressed in the endocardium based on RNAseq analysis of E9.5 embryos' endocardial and vascular endothelial cells (*Figure 3—figure supplement 2A*, *Figure 3—figure supplement 2—source data 1* ). *Prkch*[KO] did not affect Notch activation or produce any phenotype. *Prkce*[KO] caused a slight but significant reduction of NICD in the cushion endocardium at E9.5 and caused 25% heart defects at E18.5. *Prkce; Prkch* double knockout resulted in a further loss of NICD at E9.5, leading to pericardial effusion and complete lethality at E10.5. (*Figure 3C*, *Figure 3—figure supplement 2E*). No pericardial effusion or heart beating abnormalities were noted at E9.5 in these mutant embryos (*Figure 3—figure supplement 2D*), suggesting a direct action of PKCs on Notch signaling, rather than indirect action through affecting blood flow. Treatment of pregnant mice at E9.5 with a potent PKC activator, phorbol 12-myristate 13-acetate (PMA), almost completely rescued dofetilide-induced loss of NICD, pPKC, cushion hypoplasia, and cardiac defects (*Figure 3D-F*). The restricted activation of NICD in the AVC region by PMA treatment is consistent with the restricted expression of PKCε and PKC$\eta$ in the AVC endocardium (*Figure 3—figure supplement 2B*). Thus, PKC activity is both necessary and sufficient for Notch activation in the cushion endocardium.

Both PKC and AKT belong to the AGC kinase family and both have a conserved hydrophobic motif at the C-terminal tail of the catalytic domain, whose phosphorylation is dependent on mTORC2 and is required for kinase activity (*Baffi et al., 2021*). Therefore, the observed inhibition of hydrophobic motif phosphorylation of PKC and AKT indicates impairment of the common upstream activator mTORC2. We conditionally deleted one of the essential mTORC2 components, Rictor (*Sarbassov et al., 2004*), in the endothelial cells (*Figure 3—figure supplement 2C*). As expected, ablation of mTORC2 completely blocked hydrophobic motif phosphorylation of both PKC and AKT and consequently led to a significant decrease of NICD in the cushion endocardium (*Figure 3G*). Thus, blood flow activates Notch in the cushion endocardium via the mTORC2-PKC signaling pathway.

## Shear stress-induced alteration of membrane lipid microstructure activates mTORC2-PKC-Notch signaling pathway

In cultured mammalian endothelial cells, fluid shear stress (FSS) increases the number of cell surface caveolae, a specialized membrane microdomain enriched for cholesterol and sphingolipids; sequestration of cholesterol inhibits shear-dependent activation of ERK (*Park et al., 1998*). Thus, we tried to manipulate the membrane lipid microdomain by treating the cultured E9.5 embryos with cholesterol. Staining of Caveolin-1, the major component of caveolae, showed that Caveolin-1 was normally expressed on the cell surface of the AVC endocardial cells including the luminal surface and the lateral cell adhesion sites. However, in areas with low shear stress such as dorsal aorta endothelium, atrial, and ventricular endocardium downstream of AVC, Caveolin-1 was mainly restricted to the lateral cell adhesion sites and absent on the luminal surface. Ex vivo dofetilide treatment caused retraction of Caveolin-1 from the luminal surface to the lateral cell adhesion sites in the AVC endocardial cells, while co-treatment with cholesterol rescued the presence of Caveolin-1 to the luminal surface of AVC endocardial cells in the presence of dofetilide (*Figure 4A*). In addition, scanning electron microscopy on E9.5 AV canal endocardium showed numerous membrane invaginations on the luminal surface of the endocardial cells. The size of the invaginations ranged from 50 to 100 nm, consistent with the reported size of caveolae. Dofetilide significantly reduced the number of membrane invaginations, which recovered after restore of blood flow at 5 hr post dofetilide treatment (*Figure 4—figure supplement 1A*). The reduction of membrane invaginations at 3 hr post ex vivo dofetilide treatment could be rescued by co-treatment of cholesterol (*Figure 4B*). The loss of NICD, pPKC[Ser660], and pAKT[Ser473] caused by dofetilide could all be rescued by pretreatment with cholesterol, indicating the reactivation

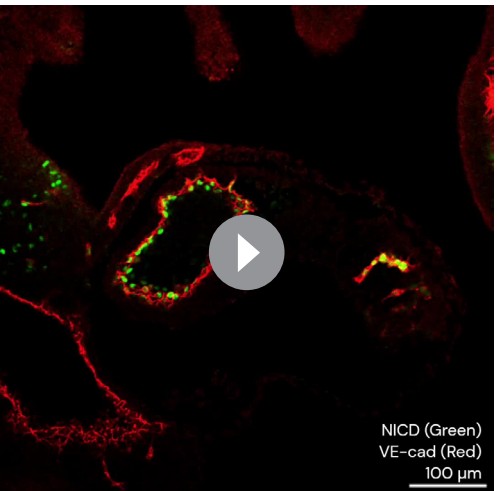

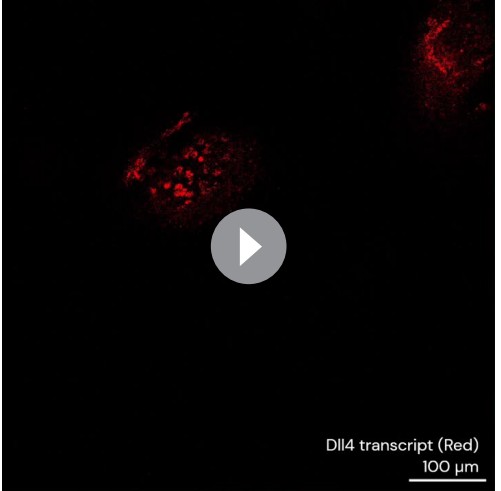

**Video 3.** E9.5 NICD whole mount staining. NICD (green) whole-mount immunofluorescence staining of E9.5 mouse heart.

https://elifesciences.org/articles/97268/figures#video3

**Video 4.** E9.5 Dll4 in situ hybridization. Whole-mount *Dll4* (red) in situ hybridization staining of E9.5 mouse heart.

https://elifesciences.org/articles/97268/figures#video4

of mTORC2 (*Figure 4C*). These rescuing effects of cholesterol disappeared in the *Rictor* knock-out embryos (*Figure 4C*), suggesting mTORC2 was intermediate signaling molecules linking membrane lipid microstructure and Notch activation.

## Pharmacogenetic interaction in the etiology of congenital heart defects

Gene-environmental interactions are believed to underlie the complex etiology of many congenital heart diseases. With the mechanosensitive nature of Notch in mind, we speculated that reducing the gene dosage of Notch1 may interact with agents that disrupt normal hemodynamic stimulations to the endocardium and synergistically cause heart defects. To test this, we crossed Notch1 heterozygous null male (FVB) with wild-type female mice, and treated the pregnant mice with a lower dose of dofetilide (1.8 mg/kg) at E9.5. In addition, we tested another drug verapamil, a commonly used FDA-approved L-type calcium channel blocker for treatment of high blood pressure, heart arrhythmias, and angina (*Fahie and Cassagnol, 2023*), in FVB background. A single dose of maternal verapamil treatment at E9.5 significantly decreased embryonic heart rate (*Figure 5E*). Neither Notch1 heterozygosity alone, nor drug treatment alone at the applied dosage produced any notable heart defects. However, the combination of Notch1 mutation and drug exposure of either dofetilide or verapamil resulted in over 50% of fetuses having various heart defects of endocardial origin (*Figure 5A–D*).

## Discussion

In this study, we identified fluid shear stress as the primary activator of Notch signaling in the area of AVC and OFT of the mouse looping heart tube. Strong fluid shear stress in the AVC and OFT enhances PKC phosphorylation by mTORC2 possibly by maintaining a particular membrane microstructure. Activated PKC then augments Notch cleavage under the minimal ligand stimulation, resulting in Notch activation and EMT specifically in areas of valve primordium characteristic of narrow internal diameter and high shear stress. These findings are directly relevant to endocardial patterning. Notch activity is known to be essential for establishing OFT and AVC endocardial cell identity capable of EMT (*Timmerman et al., 2004*; *Luna-Zurita et al., 2010*). Activation of Notch in the AVC endocardium is presumed to be mainly driven by the Dll4 ligand because Dll4 has been demonstrated to be expressed in the AVC endocardial cells and endothelial-specific knockout of *Dll4* produced acellularized AV cushions (*MacGrogan et al., 2016*). However, careful examination of the immunostaining pattern revealed that NICD, VEGFR2, and Dll4 were initially expressed with equal levels throughout cardiovascular endothelium at E8.5 prior to endocardial EMT, but at E9.5 with the onset of EMT, endocardial VEGFR2 and Dll4 were markedly reduced and NICD becomes restricted to endocardium

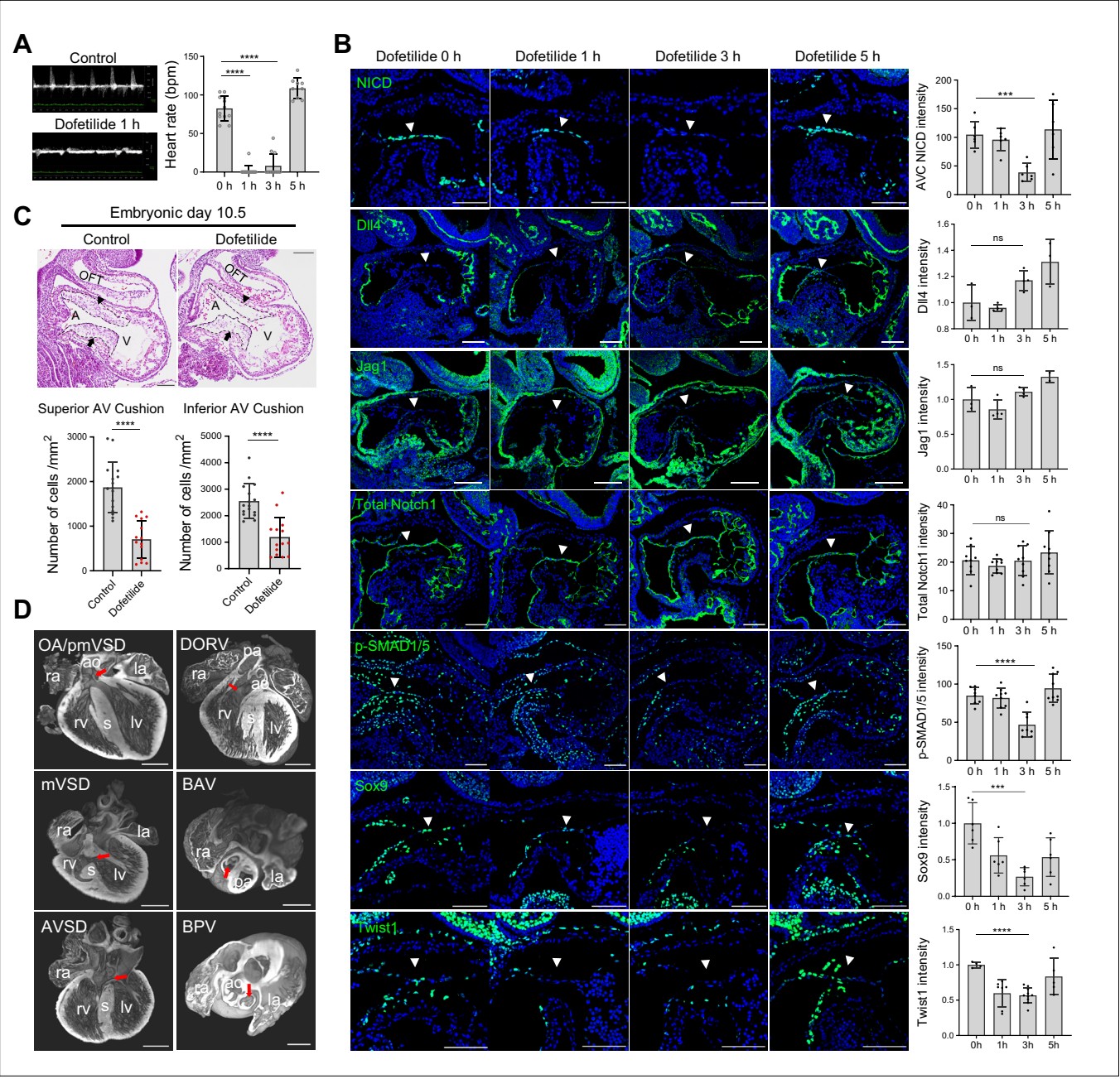

**Figure 2.** Notch activation in the cushion endocardium is dependent on blood flow. (**A**) Echocardiography of control and dofetilide treated embryos. (0 hr: n=12; 1 hr: n=15; 3 hr: n=12; 5 hr: n=9). (**B**) Expression of NICD, Dll4, Jag1, total Notch1, p-SMAD1/5, and Sox9, Twist1 in the E9.5 atrioventricular (AV) canal endocardium (arrowhead). Each point in the quantification chart represents one embryo. (**C**) Sagittal E10.5 hematoxylin-eosin stained sections demonstrated hypocellularity in both superior (arrowhead) and inferior AV cushion (arrow) caused by dofetilide treatment. Quantification of mesenchymal cell density in superior (below left) and inferior (below right) AV cushion (Control: n=16 embryos; Dofetilide: n=14 embryos). OFT, outflow tract; A, atrium; V, ventricle. (**D**) Representative heart defects induced by maternal dofetilide treatment. pmVSD, perimembranous ventricular septal defect; DORV, double-outlet right ventricle; mVSD, muscular VSD; OA, overriding aorta; BAV, bicuspid aortic valve; AVSD, atrioventricular septal defect; BPV, bicuspid pulmonary valve; ra, right atrium; la, left atrium; ao, aorta; rv, right ventricle; lv, left ventricle; s, interventricular septum; pa, pulmonary artery. Scale bars, 100 μm (**B, C**), 500 μm (**D**). Differences between groups were analyzed by t-test. Data are expressed as the mean ± SD. ***p<0.001, ****p<0.0001, ns: non-significant.

The online version of this article includes the following figure supplement(s) for figure 2:

**Figure supplement 1.** Blood flow is essential for Notch activation in the cushion endocardium and the development of the endocardial cushion.

**Figure supplement 2.** Notch inactivation in the cushion endocardium results from cardiac noncontraction rather than hypoxia.

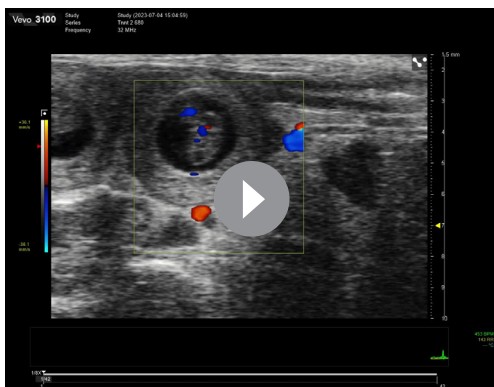

**Video 5.** E9.5 echo of a beating embryonic heart. Echocardiography of a normal E9.5 mouse embryo in utero using color Doppler.

https://elifesciences.org/articles/97268/figures#video5

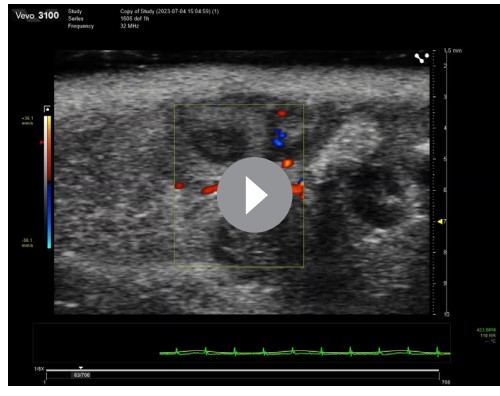

**Video 6.** E9.5 echo of a dofetilide-treated embryonic heart. Echocardiography of a 1 hr after dofetilide-treated E9.5 mouse embryo in utero using color Doppler.

https://elifesciences.org/articles/97268/figures#video6

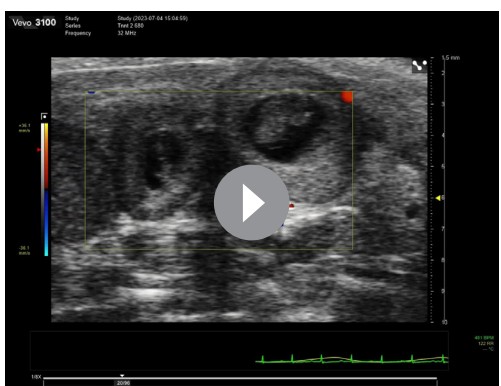

**Video 7.** E9.5 echo of *Tnnt2-cre* x *Tnnt2* flox/flox beating with non-beating. Echocardiography of two E9.5 embryos from *Tnnt2-cre; Tnnt2* flox/+ x *Tnnt2* flox/flox crossing. One embryo was genotyped to be *Tnnt2*flox/flox and the other *Tnnt2-cre; Tnnt2*flox/flox.

https://elifesciences.org/articles/97268/figures#video7

overlaying the merging OFT and AVC cushions, implying a transition of the main driver of Notch activation from the pan-endothelial VEGFR2-Dll4 stimulation at E8.5 to the regionally restricted shear force stimulation at E9.5.

We, therefore, propose the following working model as how mechanical cues guide Notch activation during endocardial patterning. Following gastrulation, a high level of VEGF-VEGFR2 signaling in mesodermal tissues initiates Dll4 expression which activates Notch and drives arterial endothelium and endocardium differentiation and proliferation. At E9.5 when endocardial EMT starts, VEGF signaling must dampen due to its suppressive function on EMT (*Dor et al., 2001*; *Chang et al., 2004*). One way to lower VEGF signaling in the endocardium is through downregulating VEGFR2 expression. Consequently, Dll4 expression also decreases, rendering Notch less active in the endocardium compared to the vascular endothelium. In the meantime, cardiac ballooning on both sides of the AVC expands the myocardial chambers and widens the endocardial internal diameter, leaving only the endocardium of the OFT and AVC endocardial cushion region closely attached. Narrow blood passageway creates strong shear stress to stimulate Notch cleavage only in these narrow areas of future cardiac septation and valve formation. Thus, in the developing mouse hearts: (1) VEGF signaling is reduced to permit endocardial EMT; (2) Dll4 expression is reduced to prevent widespread endocardial Notch activation and make endocardium sensitive to flow; (3) a proper cushion size and shape is maintained by limiting the flanking endocardium to undergo EMT despite physically close to the field of BMP2 derived from of AVC myocardium (*Figure 6*).

Studies in zebrafish identified several cation channels including Trpv4, Trpp2, Piezo1, Piezo2, and P2X that contribute to mechanosensing in the developing endocardium (*Duchemin et al., 2019*; *Heckel et al., 2015*; *Fukui et al., 2021*). Blood flow in the looping mammalian heart is predominantly unidirectional whereas early zebrafish embryonic heart displays oscillatory flow pattern (*Heckel et al., 2015*). Mammalian embryonic endocardium undergoes extensive EMT to form valve primordia while zebrafish atrioventricular valve primordia is formed via partial EMT and the collective cell migration of endocardial cells into the cardiac jelly followed by tissue sheet delamination (*Duchemin et al., 2019*; *Paolini et al., 2021*; *Chow et al., 2022*). It is thus possible that mammalian hearts may evolve

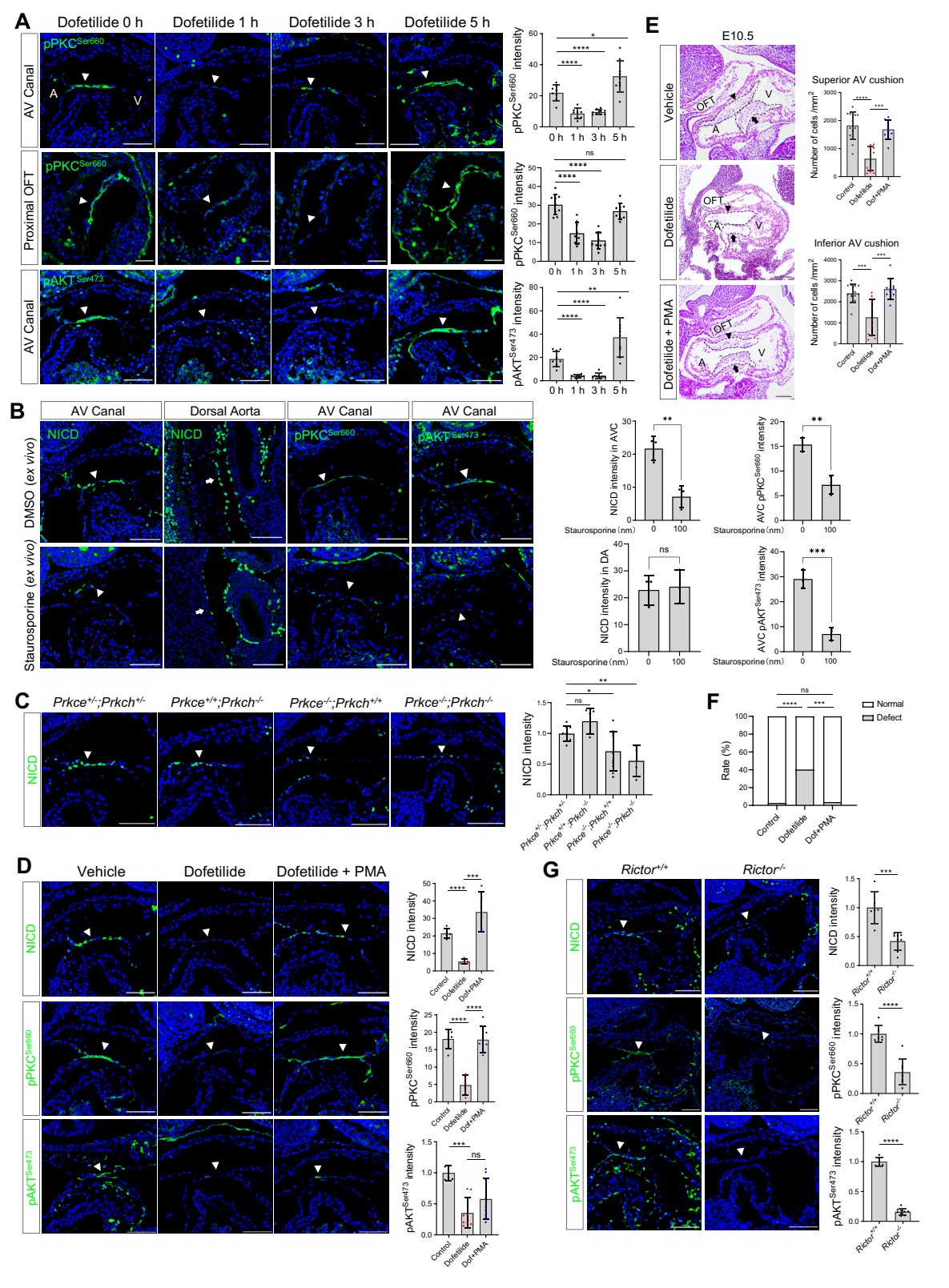

**Figure 3.** Flow-responsive mTORC2-PKCε activity is required for Notch activation in the cushion endocardium. (**A**) Phospho-PKC$^{Ser660}$ and phospho-AKT$^{Ser473}$ levels in the atrioventricular (AV) canal endocardium and proximal outflow tract (OFT) endocardium (arrowhead) after dofetilide treatment. Each point in the quantification chart represents one embryo. (**B**) NICD, phospho-PKC$^{Ser660}$ and phospho-AKT$^{Ser473}$ expression in cultured E9.5 heart in response to ex vivo staurosporine treatment (100 nM). Each point in the quantification chart represents one embryo. (**C**) NICD expression in AV canal

*Figure 3 continued on next page*

*Figure 3 continued*

endocardium in *Prkce* and *Prkch* double heterozygous, single knockout and double knockout embryos. Each point in the quantification chart represents one embryo. (**D**) NICD, phospho-PKC$^{Ser660}$ and phospho-AKT$^{Ser473}$ staining in atrioventricular canal (AVC) endocardium (arrowhead) after dofetilide treatment and after rescue by phorbol 12-myristate 13-acetate (PMA). Each point in the quantification chart represents one embryo. (**E**) Sagittal E10.5 HE staining sections demonstrate acellularized superior (arrowhead) and inferior AV cushion (arrow) caused by dofetilide which were rescued by in PMA treatment (2 mg/kg). OFT, outflow tract; A, atrium; V, ventricle. Mesenchymal cell density was quantitated in superior and inferior AV cushions (Control: n=14 embryos; Dofetilide: n=11 embryos; Dofetilide + PMA: n=9 embryos). (**F**) Heart defect rate caused by maternal dofetilide and PMA treatment (Control: n=37; Dofetilide: n=62; Dofetilide + PMA: n=27). (**G**) NICD, phospho-PKC$^{Ser660}$, and phospho-AKT$^{Ser473}$ in AV canal endocardium (arrowhead) in wild-type and *Rictor* null mice (*Rictor*$^{+/+}$: n=6; *Rictor*$^{-/-}$: n=8). Scale bars, 100 μm. Differences between groups were analyzed by t-test (**A-E, G**) and Two-sided Fisher's exact test (**F**). Data are expressed as the mean ± SD. *$p<0.05$, **$p<0.01$, ***$p<0.001$, ****$p<0.0001$, ns: non-significant.

The online version of this article includes the following source data and figure supplement(s) for figure 3:

**Figure supplement 1.** phospho-PKC$^{Ser660}$ and phospho-AKT$^{Ser473}$ are undectable in E9.5 dorsal aorta endothelium.

**Figure supplement 2.** PKCε and PKC $\eta$ are the most abundantly expressed PKC isoforms in the endocardium.

**Figure supplement 2—source data 1.** RNA sequencing data of E9.5 embryonic endocardial cells.

additional mechanisms to guide endocardial patterning and development. Our data support a model that the membrane lipid microdomain acts as a shear stress sensor and transduces the mechanical cue to activate the intracellular mTORC2-PKC-Notch signaling pathway in the developing endocardium. Shear stress has been shown to upregulate the number of caveolae in cultured endocardial cells *Park et al., 1998*. Flow-induced AKT phosphorylation is dependent on Caveolin-1 (*Albinsson et al., 2008*). Both PKCα and PKCε interact with Caveolin-1 and bind with caveolae membranes (*Mineo et al., 1998*; *Oka et al., 1997*). In neuronal cells or neural tissues, accumulation of psychosine in the plasma membrane disrupted lipid rafts, blocked recruitment of mTORC2 and PKC to the lipid raft, and inhibited AKT and PKC activation (*Sural-Fehr et al., 2019*; *White et al., 2009*). In line with these previous findings, we demonstrated that high level of shear stress in the AVC and OFT is required for caveolin localization to the luminal surface of the endocardial cell membrane, and this membrane lipid microstructure is both necessary and sufficient for mTORC2-PKC-Notch pathway activation. The mechanism by which shear stress and cholesterol increases caveolae is unknown. In a previous study, shear stress exposure for a few minutes rapidly decreases the lipid order and increases the fluidity of the plasma membranes which appears to contradict our findings (*Yamamoto and Ando, 2013*). However, membrane lipid compositions are highly dynamic and regulated. It is possible that acute shear stress and its associated kinetic energy may initially decrease membrane lipid order, but long-term shear may evoke cellular feedback mechanisms to increase lipid order and enhance membrane rigidity. As cholesterol is an integral component of lipid raft and caveolae, it is likely that enrichment of cholesterol to the plasma membrane by exogenous supplementation might alter the membrane structure to make the lipid raft structure less dependent on sheer stress. It is noteworthy that the methodology used to alter blood flow in this study inevitably affects myocardial contraction. Thus, further work to uncouple changes in shear stress and myocardial mechanical properties, with the aid of theoretical modeling or using mouse heart valve explants, is needed to fully characterize the effect of shear stress on mouse endocardial development.

The positive regulation of Notch activation by PKC has been reported in a number of in vitro systems. Studies in CD4$^+$ T cells revealed that Notch is activated within hours of TCR stimulation independent of Notch ligands, and PKC activity is both sufficient and necessary for the ligand-independent Notch activation (*Steinbuck et al., 2018*). The exact mechanism by which PKC regulates Notch processing is not known, and in need of further investigation in the future.

The vast majority of congenital heart diseases do not have a genetic explanation and are considered to have a multifactorial origin. Studies conducted on various model organisms such as chick and zebrafish have demonstrated that alterations in blood flow during the early stages of heart development can lead to heart malformation (*Goenezen et al., 2012*). Similarly, human studies have shown that fetal bradycardia or abnormal flow patterns in the Ductus Venosus during the first trimester of pregnancy are associated with an increased risk of CHD (*Doubilet et al., 1999*; *Braga et al., 2019*). Our research findings suggest that abnormal heart contraction or flow patterns, resulting from genetic mutations or the use of certain drugs during early heart development, may interact with genetic pathways involved in mechanosensing and endocardial EMT. These interactions could contribute to the complex etiology of congenital heart disease.

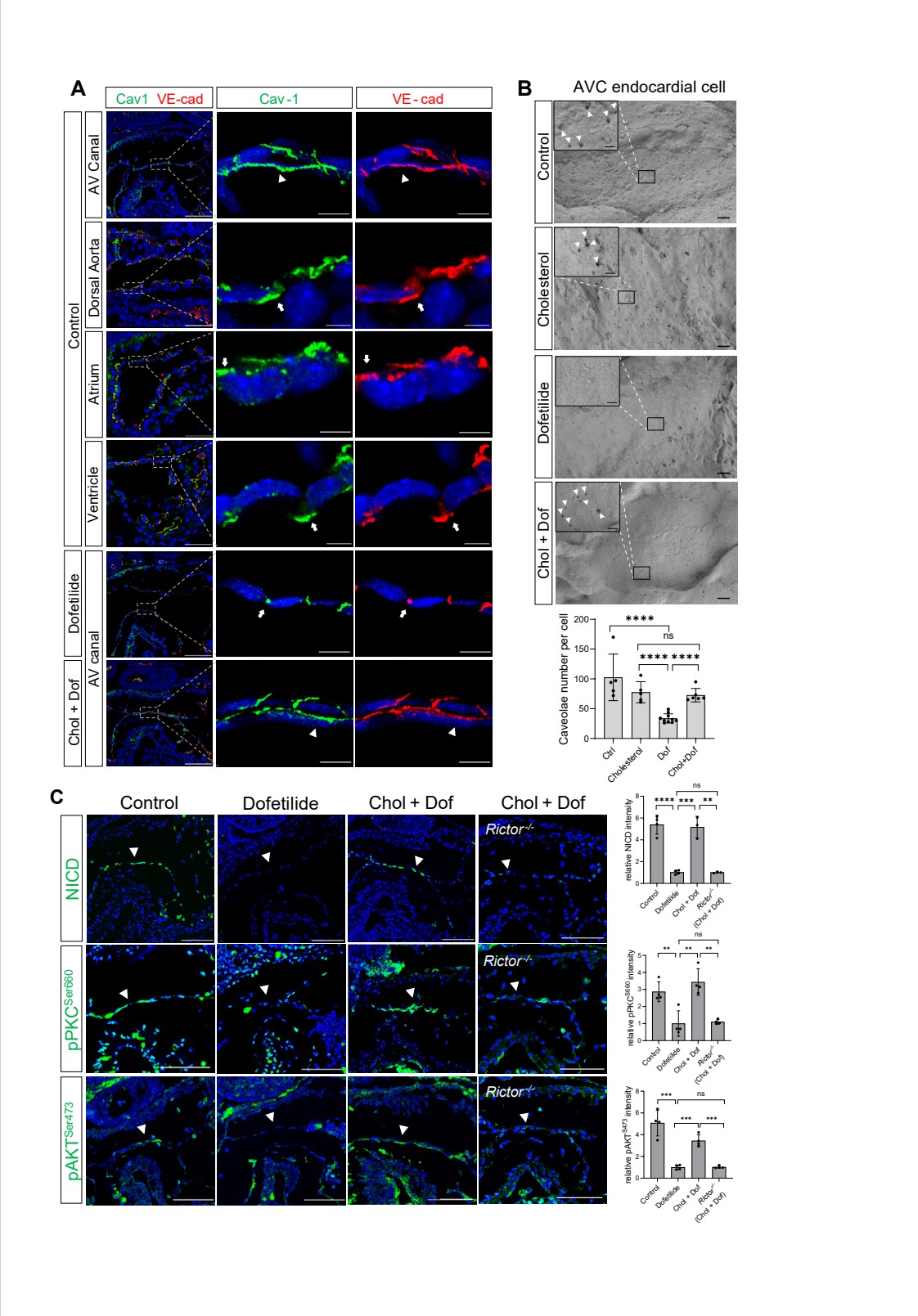

**Figure 4.** Shear stress-induced alteration of membrane lipid microstructure activated mTORC2-PKC-Notch signaling pathway. (**A**) Caveolin-1 and VE-cadherin expression in mouse E9.5 atrioventricular (AV) canal, dorsal aorta, atrium, and ventricle endocardium, demonstrating luminal (arrowhead) and lateral (arrow) surface localization. Ex vivo dofetilide treatment (0.2 μg/ml) of cultured E9.5 embryos for 3 hr caused retraction of Caveolin-1 and VE-cadherin from the luminal surface of the atrioventricular canal (AVC) endocardial cells to the lateral cell adhesion sites which could be rescued by co-

*Figure 4 continued on next page*

*Figure 4 continued*

treatment with cholesterol (1 mg/ml). (**B**) Representative images and quantifications of scanning electron microscopy on E9.5 embryonic heart AV canal endocardial cells at their luminal surfaces, with caveolae structure presentively pointed by arrowhead. Each point in the quantification chart represents one embryo. (**C**) Loss of NICD, phospho-PKC$^{Ser660}$, and phospho-AKT$^{Ser473}$ in AVC endocardium (arrowhead) by ex vivo dofetilide treatment could be rescued by cholesterol. The rescue failed in *Rictor* null hearts. Each point in the quantification chart represents one embryo. Scale bars, 100 μm (**A, C**), 10 μm (**A**, zoom-in), 1 μm (**B**), 200nm (**B**, zoom-in). Differences between groups were analyzed by t-test. Data are expressed as the mean ± SD. **p<0.01, ***p<0.001, ****p<0.0001, ns: non-significant.

The online version of this article includes the following figure supplement(s) for figure 4:

**Figure supplement 1.** Dofetilide treatment reduced the caveolae structure in atrioventricular (AV) canal endocardial cells.

# Materials and methods

## Key resources table

| Reagent type (species) or resource | Designation | Source or reference | Identifiers | Additional information |
|---|---|---|---|---|
| strain, strain background (*M. musculus*) | Dll4$^{flox/flox}$ | Cyagen | C57BL/6JCya-Dll4$^{em1flox}$/Cya | |
| strain, strain background (*M. musculus*) | Tek-cre | GemPharmatech | C57BL/6JGpt-H11$^{em1Cin(Tek-iCre)}$/Gpt | RRID:IMSR_GPT:T003764 |
| strain, strain background (*M. musculus*) | Tnnt2$^{flox/flox}$ | GemPharmatech | C57BL/6JGpt-Tnnt2$^{em1Cflox}$/Gpt | RRID:IMSR_GPT:T013227 |
| strain, strain background (*M. musculus*) | Tnnt2-cre | Jax | STOCK Tg(Tnnt2-cre)5Blh/JiaoJ | RRID:IMSR_JAX:024240 |
| strain, strain background (*M. musculus*) | Prkce$^{KO}$ | This paper | | The mouse line was generated in house using a sgRNA with the following target site GGAAGCGGCAAGGGGCTGTC. The sgRNA-cas9 ribonucleoprotein was injected to the zygotes to produce the mouse line with the following null allele NC_000083.7:g.86781801_86781811del. |
| strain, strain background (*M. musculus*) | Prkch$^{KO}$ | This paper | | Mouse line generated in house using a sgRNA with the following target site TCAAGTGAACGGACATAAGT. The sgRNA-cas9 ribonucleoprotein was injected to the zygotes to produce the mouse line with the following null allele NC_000078.7:g.73738396_73738411del. |
| strain, strain background (*M. musculus*) | Rictor$^{KO}$ | This paper | | Mouse line generated in house using a sgRNA with the following target site GCCAACTCATTAATTGCGGT. The sgRNA-cas9 ribonucleoprotein was injected to the zygotes to produce the mouse line with the following null allele NC_000081.7:g.6785997_6785998ins AAACAGGTCAATTAATT. |
| strain, strain background (*M. musculus*) | Notch1$^{KO}$ | This paper | | Mouse line generated in house using a sgRNA with the following target site ATGTCTGTCAACAGCTGCAG. The sgRNA-cas9 ribonucleoprotein was injected to the zygotes to produce the mouse line with the following null allele NC_000068.8:g.26375459_26375768delinsTTGGG. |
| strain, strain background (*M. musculus*) | Epor$^{P2A-icre}$ | GemPharmatech | C57BL/6JGpt-Epor$^{em1Cin(P2A-iCre-P2A-EGFP)}$/Gpt | RRID:IMSR_GPT:T052749 |
| strain, strain background (*M. musculus*) | ROSA-DTA | Jax | B6.129P2-Gt(ROSA)26Sor$^{tm1(DTA)Lky/J}$ | RRID:IMSR_JAX:009669 |
| strain, strain background (*M. musculus*) | FVB | Beijing HFK | | |
| strain, strain background (*M. musculus*) | C57bl/6 | Shanghai Jihui Laboratory | | |

*Continued on next page*

*Continued*

| Reagent type (species) or resource | Designation | Source or reference | Identifiers | Additional information |
|---|---|---|---|---|
| antibody | Cleaved Notch (NICD) (Rabbit monoclonal) | Cell Signaling Technology | Cat# 4147, RRID:AB_2153348 | IF(1:400, TSA) |
| antibody | Phospho-AKT (Ser473) (D9E) (Rabbit monoclonal) | Cell Signaling Technology | Cat# 4060, RRID:AB_2315049 | IF(1:400, TSA) |
| antibody | Phospho-SMAD1/5 (Ser463/465) (41D10)(Rabbit monoclonal) | Cell Signaling Technology | Cat# 9516, RRID:AB_491015 | IF(1:400, TSA) |
| antibody | Phospho-PKC (pan) (β II Ser660) (Rabbit monoclonal) | Cell Signaling Technology | Cat# 9371, RRID:AB_2168219 | IF(1:400, TSA) |
| antibody | Jagged1 (D4Y1R)(Rabbit monoclonal) | Cell Signaling Technology | Cat# 70109, RRID:AB_2799774 | IF(1:400, TSA) |
| antibody | VEGF receptor 2 (D5B1) (Rabbit monoclonal) | Cell Signaling Technology | Cat# 9698, RRID:AB_11178792 | IF(1:400) |
| antibody | PKCε (22B10)(Rabbit monoclonal) | Cell Signaling Technology | Cat# 2683, RRID:AB_2171906 | IF(1:400, TSA) |
| antibody | PKC eta (EPR18513)(Rabbit monoclonal) | Abcam | Cat# ab179524, RRID:AB_2892155 | IF(1:400, TSA) |
| antibody | Notch1 (EP1238Y)(Rabbit monoclonal) | Abcam | Cat# ab52627, RRID:AB_881725 | IF(1:400, TSA) |
| antibody | VE-Cadherin (Goat polyclonal) | R&D Systems | Cat# AF1002, RRID:AB_2077789 | IF(1:400) |
| antibody | Dll4 (Goat polyclona) | R&D Systems | Cat# AF1389, RRID:AB_354770 | IF(1:200) |
| antibody | Troponin T (RV-C2) (Rabbit monoclonal) | DSHB | Cat# RV-C2, RRID:AB_2240831 | IF(1:200) |
| antibody | Sox9 (SN74-09) (Rabbit monoclonal) | HuaBio | Cat# ET1611-56, RRID:AB_2924312 | IF(1:200) |
| antibody | Twist1 (Mouse monoclonal) | Abcam | Cat# ab50887, RRID:AB_883294 | IF(1:200) |
| antibody | HRP-Donkey Anti-Rabbit IgG (H+L) (Donkey polyclonal) | Jackson ImmunoResearch | Cat# 711-035-152, RRID:AB_10015282 | IF(1:500) |
| antibody | Peroxidase AffiniPure Donkey Anti-Mouse IgG (H+L) (Donkey polyclonal) | Jackson ImmunoResearch | Cat# 715-035-150, RRID:AB_2340770 | IF(1:500) |
| antibody | Peroxidase AffiniPure Donkey Anti-Goat IgG (H+L) (Donkey polyclonal) | Jackson ImmunoResearch | Cat# 705-035-003, RRID:AB_2340390 | IF(1:500) |
| antibody | Cy5 AffiniPure Donkey Anti-Goat IgG (H+L) (Donkey polyclonal) | Jackson ImmunoResearch | Cat# 705-175-147, RRID:AB_2340415 | IF(1:500) |
| antibody | Cy3 AffiniPure Donkey Anti-Goat IgG (H+L) (Donkey polyclonal) | Jackson ImmunoResearch | Cat# 711-165-152, RRID:AB_2307443 | IF(1:500) |
| antibody | Alexa Fluor 488 AffiniPure Donkey Anti-Mouse IgG (H+L) (Donkey polyclonal) | Jackson ImmunoResearch | Cat# 715-545-150, RRID:AB_2340846 | IF(1:500) |
| antibody | Cy3 Streptavidin | Jackson ImmunoResearch | Cat# 016-160-084, RRID:AB_2337244 | IF(1:500) |
| antibody | Alexa Fluor 488 Streptavidin | Jackson ImmunoResearch | Cat# 016-540-084, RRID:AB_2337249 | IF(1:500) |
| antibody | DAPI | Beyotime | Cat# C1002 | IF(1:500) |
| sequence-based reagent | Dll4flox/flox _F | This paper | PCR primers | CAACTGACCTAAAATGGGATGGTG |
| sequence-based reagent | Dll4flox/flox _R | This paper | PCR primers | GGTAACTACAAGGCAGAAAGAGGA |
| sequence-based reagent | Tnnt2flox/flox _F | This paper | PCR primers | GATCCTGCCTCCTTAGGTCTCAAGT |
| sequence-based reagent | Tnnt2flox/flox _R | This paper | PCR primers | CAAGTTCCTATGCCACATCTGCATG |
| sequence-based reagent | Tek-Cre _F | This paper | PCR primers | GCGGTCTGGCAGTAAAAACTATC |
| sequence-based reagent | Tek-Cre _R | This paper | PCR primers | GTGAAACAGCATTGCTGTCACTT |

*Continued on next page*

*Continued*

| Reagent type (species) or resource | Designation | Source or reference | Identifiers | Additional information |
|---|---|---|---|---|
| sequence-based reagent | Tnnt2-cre_F | This paper | PCR primers | GGACATGTTCAGGGATCGCCAGGCG |
| sequence-based reagent | Tnnt2-cre_R | This paper | PCR primers | GCATAACCAGTGAAACAGCATTGCTG |
| sequence-based reagent | Prkce$^{KO}$_F | This paper | PCR primers | GAGTGTTCAGGGAGCGTATG |
| sequence-based reagent | Prkce$^{KO}$_R | This paper | PCR primers | CAAGTAGGTGGCCATGAACTTG |
| sequence-based reagent | Prkch$^{KO}$_F | This paper | PCR primers | GAGACCGCATCTTCAAGC |
| sequence-based reagent | Prkch$^{KO}$_R | This paper | PCR primers | GTAGGTGGGCTGCCTC |
| sequence-based reagent | Rictor$^{KO}$_F | This paper | PCR primers | TCCTTCTCTGTTACAGATG |
| sequence-based reagent | Rictor$^{KO}$_R | This paper | PCR primers | ACCATTCTGTCTCGTTC |
| sequence-based reagent | Rosa-DTA_F | This paper | PCR primers | GTTATCAGTAAGGGAGCTGCAGTGG |
| sequence-based reagent | Rosa-DTA_wt_R | This paper | PCR primers | GGCGGATCACAAGCAATAATAACC |
| sequence-based reagent | Rosa-DTA_mt_R | This paper | PCR primers | AAGACCGCGAAGAGTTTGTCCTC |
| sequence-based reagent | Epor$^{P2A-icre}$_F | This paper | PCR primers | TCCCACTCCACCTCACTTGAAG |
| sequence-based reagent | Epor$^{P2A-icre}$_R | This paper | PCR primers | CTGACTTCATCAGAGGTGGCATC |
| sequence-based reagent | Notch1$^{KO}$_F | This paper | PCR primers | AAGCTGGGAGAGAAAAGCAGACC |
| sequence-based reagent | Notch1$^{KO}$_R | This paper | PCR primers | CACAACCTCCTATAGCCCTTACC |
| chemical compound, drug | Staruosporine | MCE | HY-15141 | |
| chemical compound, drug | Phorbol 12-myristate 13-acetate (PMA) | Sigma-Aldrich | P8139 | |
| chemical compound, drug | Dofetilide | Rhawn | R023594 | |
| chemical compound, drug | Wortmannin | Selleck | S2758 | |
| chemical compound, drug | Cholesterol-Water soluble | Sigma-Aldrich | C4951 | |
| chemical compound, drug | CMC-Na | Shyuanye | S14016 | |
| chemical compound, drug | Saline | Shyuanye | R27405 | |
| chemical compound, drug | Isoflurane | RWD | R510-22-10 | |
| chemical compound, drug | 10% neutral buffered formalin | Legene | DF0111 | |
| chemical compound, drug | Glutaraldehyde | Sigma-Aldrich | G7651 | |
| chemical compound, drug | Benzyl alcohol | Sinoreagent | 30020618 | |
| chemical compound, drug | Benzyl benzoate | Sinoreagent | 30020828 | |
| chemical compound, drug | Glycerol | Sinoreagent | 10010618 | |

*Continued on next page*

*Continued*

| Reagent type (species) or resource | Designation | Source or reference | Identifiers | Additional information |
|---|---|---|---|---|
| chemical compound, drug | Hanks buffer | Beyotime | C0218 | |
| chemical compound, drug | DMEM | Gibco | C11965500BT | |
| chemical compound, drug | DMSO | Macklin | D806645 | |
| chemical compound, drug | Triton X-100 | Macklin | I997471 | |
| chemical compound, drug | Fetal bovine sreum | Solarbio | S9030 | |
| chemical compound, drug | Verapamil | Selleck | S4202 | |
| software, algorithm | ImageJ 1.52i | ImageJ | RRID:SCR_003070 | |
| software, algorithm | Zen 2.3 | Zeiss | RRID:SCR_013672 | |
| software, algorithm | Imaris v9.1.1 | Oxford Instruments | RRID:SCR_007370 | |
| software, algorithm | Adobe Photoshop cc2019 | Adobe | RRID:SCR_014199 | |
| software, algorithm | Olympus OlyVIA 3.3 | Olympus | RRID:SCR_016167 | |
| software, algorithm | GraphPad Prism 8 | GraphPad | RRID:SCR_002798 | |
| software, algorithm | Vevo Lab 5.7.1 | FUJIFILM | RRID:SCR_022152 | |

## Reagents

A list of the reagents used in this study is provided in Key resources table.

## Animals

All mice used in this study were housed at a constant temperature (23 °C) and humidity (~50%) with a 12 h light/dark cycle and ad libitum access to food and water. All animal experiments were performed in accordance with the protocol approved by the Westlake University Institutional Animal Care and Use Committee (approval # 21–007-SHJ). Noon of the day of vaginal plug detection was defined as embryonic day (E) 0.5. For dofetilide treatment, mice were gavaged at E9.5 with a single dose of dofetilide at 2 mg/kg dissolved in saline. For phorbol 12-myristate 13-acetate (PMA, 2 mg/kg) administration, mice were gavaged at E9.5 with a single dose of PMA dissolved in CMC-Na. Verapamil was given by intraperitoneal injection at E9.5. At the indicated date post-treatment, the dams were sacrificed by cervical dislocation, and the embryos were harvested for further analysis. All sources of our mouse lines are listed in Key resources table. Knockout mice generated in-house are described and validated in *Figure 3—figure supplement 2E*.

## Fetal echocardiography

Ultrasound scanning was performed using the Vevo 300 high-frequency ultrasound machine (FUJI FILM Visual Sonics Inc, Canada). In utero echocardiography of fetal mice was conducted at E9.5. Transducer MX-550D had a central frequency of 40 MHz with axial and lateral resolutions both about 30 μm. The pregnant mice were sedated using 2% isoflurane and kept at 1% isoflurane to maintain body temperature at 35.5–37°C and heart rate at 400–500 beats per minute (bpm). Pulse-wave Doppler was used to measure velocity and time interval parameters to measure the blood flow velocity and time interval parameters of the fetal hearts. All fetuses were scanned twice in order and their in utero positions were noted. For embryos of *Tnnt2-cre* x *Tnnt2* *flox/flox*, after completing the Doppler ultrasound imaging, the embryos were dissected out with the recorded positions, and embryonic tissues were collected for genotype identification to correlate with the heart rate data. During the measurements, observers were blinded to the assigned groups. Analysis was performed with Vevo Lab 5.7.1.

## Fetal heart morphology

Fetal hearts were imaged using the Zeiss Lightsheet Z.1 microscope. Embryos were harvested at E18.5, and hearts were dissected in phosphate-buffered saline (PBS), fixed overnight in a mixed

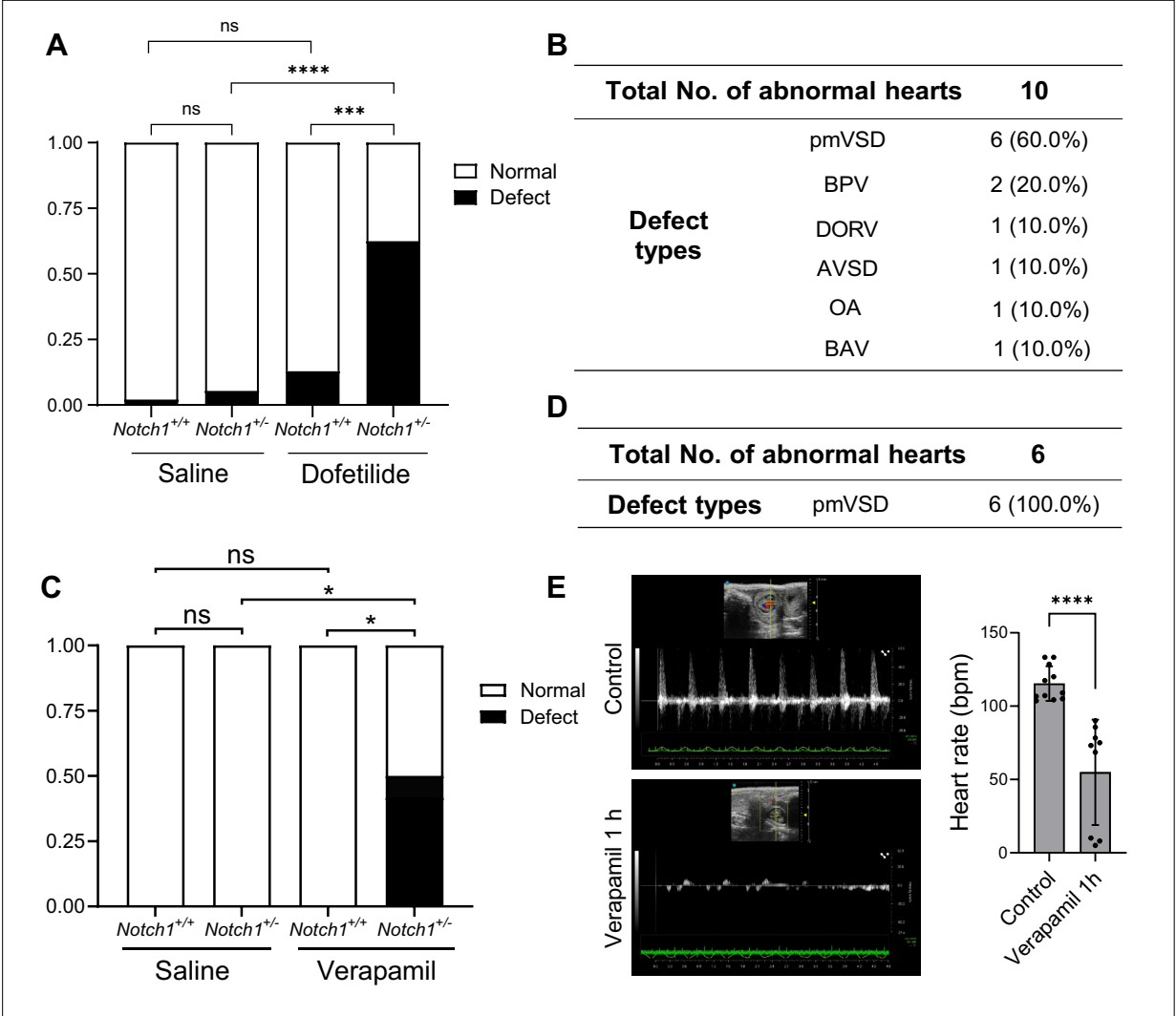

**Figure 5.** Pharmacogenetic interaction causing heart defects. (**A**) Heart defect rate significantly increased in Notch1 heterozygous embryos treated with dofetilide (1.8 mg/kg) at E9.5. (Saline *Notch1*$^{+/+}$: *n*=48, Saline *Notch1*$^{+/-}$: *n*=37, Dofetilide *Notch1*$^{+/+}$: *n*=31, Dofetilide *Notch1*$^{+/-}$: *n*=16). Differences between groups were analyzed by Two-sided Fisher's exact test. (**B**) Types of heart defects in the dofetilide and *Notch1*$^{+/-}$ combined group. (**C**) Heart defect rate significantly increased in Notch1 heterozygous embryos treated with Verapamil (15 mg/kg) at E9.5. (Saline *Notch1*$^{+/+}$: *n* = 18, Saline *Notch1*$^{+/-}$: *n* = 9, Verapamil *Notch1*$^{+/+}$: *n* = 8, Verapamil *Notch1*$^{+/-}$: *n* = 12). Differences between groups were analyzed by Two-sided Fisher's exact test. (**D**) Type of heart defects in the verapamil combined with *Notch1*$^{+/-}$ group. (**E**) Representative echocardiography and quantifications of heartbeat in control and verapamil treated E9.5 embryos. Each point in the quantification chart represents one embryo. Differences between groups were analyzed by t-test. Data are expressed as the mean ± SD. *p<0.05, ***p<0.001, ****p<0.0001, ns: non-significant.

solution of 10% neutral buffered formalin and 2.5% glutaraldehyde, then rinsed twice in PBS, and dehydrated in 50, 75, and 100% ethanol for 30 min each at room temperature (RT). Samples were then transferred into a specially designed glass tube containing 100 µL of BABB solution (1:2 benzyl alcohol: benzyl benzoate) and incubated for 30 min to clear the sample. The glass tube was mounted into the sample chamber which was filled with 87% glycerol (RI~1.45). Hearts were scanned from the apex to the great arteries for tissue autofluorescence using the 561 nm laser line and the detection optics 5 x/0.16 (n=1.45). 3D reconstruction of the image stacks and morphological analyses were performed with Imaris 9.3 software.

## Ex vivo embryo culture

Mouse embryos at E9.5 were carefully dissected in Hanks Buffer containing calcium and magnesium to remove decidua without damaging their yolk sacs and placentas. They were then cultured in 1 mL

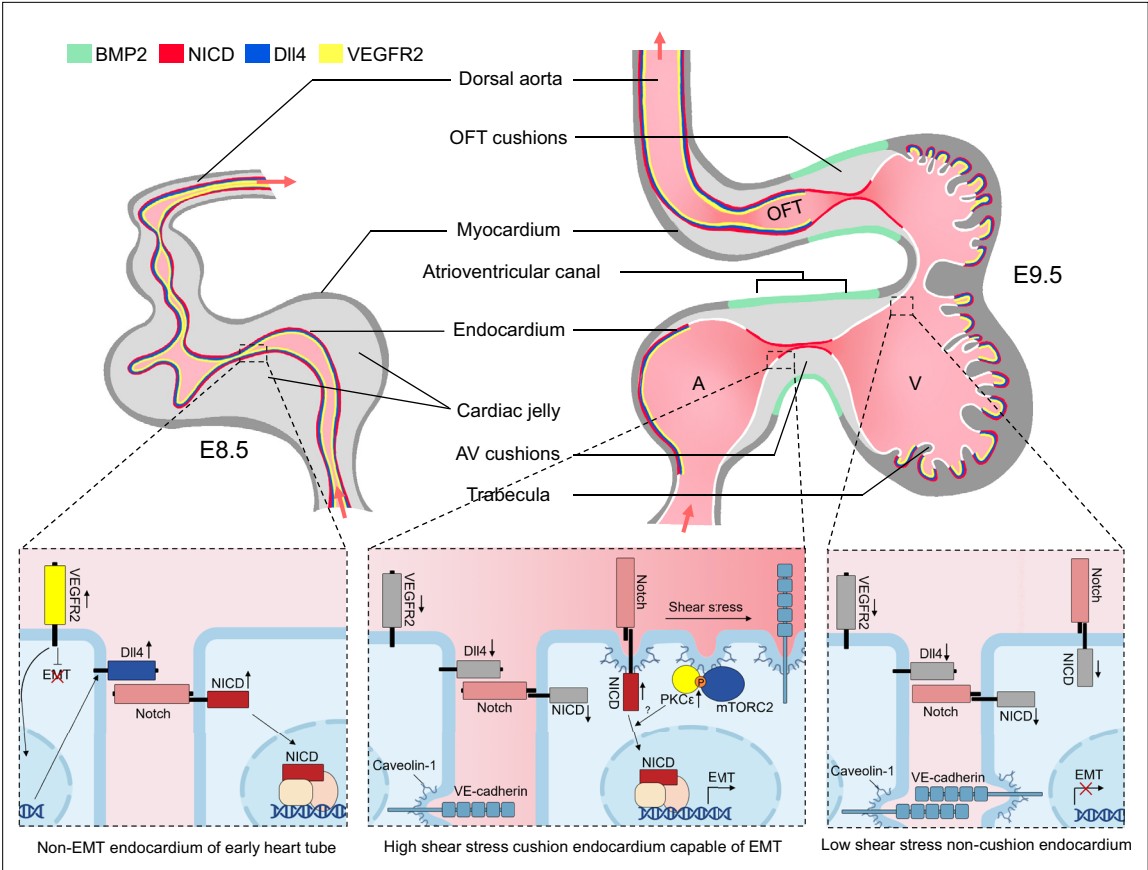

**Figure 6.** Working model of the establishment of Notch activation pattern by mechanical cues. The establishment of a Notch activation pattern by mechanical cues involves a series of events in the developing heart tube. At E8.5, the arterial endothelium and non-EMT endocardium exhibit low shear stress, high VEGF, Dll4, and Notch signaling. One day later, the endocardium undergoes patterning and becomes capable of epithelial-to-mesenchymal transition (EMT) only in the atrioventricular canal (AVC) and proximal outflow tract (OFT) regions. This patterning is achieved through restricted expression of BMP2 and NICD in these specific areas. EMT requires the downregulation of VEGF signaling by the endocardium, enabling EMT to occur. Additionally, Dll4 is downregulated in the endocardium to prevent widespread Notch activation. Simultaneously, the high shear stress present in the AVC and proximal OFT regions leads to increased membrane lipid order, which activates the mTORC2-PKC-Notch pathway and promotes EMT. On the other hand, regions flanking these valve-forming areas experience lower shear stress, resulting in inactive Notch signaling and an inability to undergo EMT.

embryo culture medium (50% rat serum, 50% DMEM), pre-saturated with a gas mix of 95% $O_2$ and 5% $CO_2$ in a 5 mL volume glass tubes. Tubes were sealed and placed in a rotatory shaker maintained at 37 °C and 50 rpm. For pharmacological treatment, stauporine (100 nM), wortmannin treatment (2 µM), dofetilide (0.2 ug/ml), or water-soluble cholesterol (1 mg/ml) was diluted in embryo culture medium to the desired concentrations.

## Immunofluorescence and in situ hybridization

E9.5 and E10.5 wild embryos were dissected in PBS, fixed in 4% paraformaldehyde (PFA) overnight at 4 °C, dehydrated in ethanol series, cleared in xylene, and embedded in paraffin. Tissues were sectioned at 6 µm thickness. Paraffin sections were subjected to a 20- min heat-induced antigen retrieval using a citric acid solution (pH 6). The sections were blocked with 5% normal donkey serum (NDS) in Tris-buffered saline with 1% Tween-20 (TBST) for 1 hr, followed by overnight incubation at 4 °C with primary antibodies diluted in the blocking buffer. Bound primary antibodies were detected using fluorescently conjugated corresponding secondary antibodies. For the detection of low-abundance targets such as NICD and phospho-PKC$^{Ser660}$, tyramide signal amplification (TSA) was applied. A list of the antibodies used in this study is provided in Key resources table. Confocal microscopy images were captured on a Zeiss LSM 800 and image analysis was performed using Zen 2.3. Fluorescent intensity was calculated using ImageJ software.

To analyze NICD expression in E8.5, E9.5 mouse hearts, whole mouse embryos were fixed in 4% PFA at 4 °C overnight. After blocking in 5% NDS, 0.5% Triton X-100 in TBS at 4 °C overnight, samples were incubated at 4 °C overnight with the primary NICD antibody diluted in blocking buffer (1:400). After washes, embryos were incubated with donkey anti-rabbit HRP (1:500) 4 °C overnight. Embryos were then washed and reacted with a tyramide-biotin solution at RT for 30 min. After that, samples were washed and incubated with Cy3 streptavidin (1:500). Stained embryos were dehydrated in methanol, cleared in BABB, and imaged by Zeiss LSM 800.

Whole-mount in situ hybridization of *Dll4* was performed using the HCR probe set according to the manufacturer's instruction (Molecular Instruments, Los Angeles). Stained embryos were dehydrated in methanol, cleared in BABB, and imaged by Zeiss LSM 800.

## Scanning electron microscopy

E9.5 embryos are fixed with a solution containing 2% paraformaldehyde and 2.5% glutaraldehyde pH 7.2 at 4 °C overnight. The fixed embryos are then washed three times with 0.1 M phosphate buffer (PB, pH 7.2–7.4) at 4 °C for 10 min each. Subsequently, the samples are further fixed with 1% osmium tetroxide in 0.1 M PB on ice for 1 hr. After fixation, they are rinsed three times with double-distilled water (ddH$_2$O) at room temperature for 15 min each. Dehydration is carried out at room temperature using a graded ethanol series: 30% ethanol for 10 min, 50% ethanol for 10 min, 70% ethanol for 10 min, 95% ethanol for 10 min, and absolute ethanol three times for 10 min each. Then the dehydrated samples are subjected to critical point drying. Samples were attached to a sample holder with double-sided carbon tape, and coated with a 10–15 nm thick metal film for SEM imaging.

## Statistics

Differences in expression levels was tested using a two-tailed t-test. Two-sided Fisher's exact test was used to compare the heart defect rates between the two groups. p-value <0.05 was considered significant. All statistical analyses were performed in GraphPad Prism 8 software.

## Acknowledgements

The authors are grateful to Dr. Xu Li of Westlake University for valuable suggestions regarding Notch experiments. We thank Dr. Zhen Zhang from Shanghai Jiao Tong University, Dr. Bing Zhang, Dr. Jiemin Jia, Dr. Shang Cai from Westlake University, and Dr Hui Zhang from Shanghai Tech University for sharing mouse lines. We thank the Microscopy Core Facility of Westlake University for assistance and advice. We thank the Westlake Animal Facility and Youshi Chen for mouse breeding and husbandry. We thank Jianfeng Wang for their assistance in ex vivo culture. This work was supported by the Natural Science Foundation of Zhejiang Province of China (LZ19H040001) and the Westlake Education Foundation.

## Additional information

### Funding

| Funder | Grant reference number | Author |
| --- | --- | --- |
| Natural Science Foundation of Zhejiang Province | LZ19H040001 | Hongjun Shi |

The funders had no role in study design, data collection and interpretation, or the decision to submit the work for publication.

### Author contributions

Yunfei Mu, Data curation, Formal analysis, Investigation, Visualization, Methodology, Writing - original draft; Shijia Hu, Data curation, Formal analysis, Investigation, Visualization, Writing - original draft; Xiangyang Liu, Methodology; Xin Tang, Jiayi Lin, Data curation; Hongjun Shi, Conceptualization, Supervision, Funding acquisition, Writing – review and editing

## Author ORCIDs
Yunfei Mu (iD) http://orcid.org/0009-0003-3749-3718
Hongjun Shi (iD) https://orcid.org/0000-0002-4993-2322

## Ethics
All animal experiments were performed in accordance with the protocol approved by the Westlake University Institutional Animal Care and Use Committee (approval # 21-007-SHJ).

Reviewer #2 (Public review): https://doi.org/10.7554/eLife.97268.3.sa1
Reviewer #3 (Public review): https://doi.org/10.7554/eLife.97268.3.sa2
Author response https://doi.org/10.7554/eLife.97268.3.sa3

---

# Additional files

## Supplementary files
MDAR checklist

## Data availability
All data generated or analysed during this study are included in the manuscript and supporting files.

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
