## [Editor Report · eLife Assessment]

Notch1 is expressed uniformly throughout the mouse endocardium during the initial stages of heart valve formation, yet it remains unclear how Notch signaling is activated specifically in the AVC region to induce valve formation. To answer this question, the authors used a combination of in vivo and ex vivo experiments in mice to demonstrate ligand-independent activation of Notch1 by circulation induced-mechanical stress and provide evidence for stimulation of a novel mechanotransduction pathway involving post-translational modification of mTORC2 and Protein Kinase C (PKC) upstream of Notch1. These findings represent an **important** advance in our understanding of valve formation and the conclusions are supported by **convincing** data.

---

## [Referee Report · Reviewer #2 (Public review)]

Summary:

In mice, Notch1 is expressed uniformly throughout the endocardium during the initial stages of heart valve formation. How, then, is Notch activated specifically in the valve forming regions? To answer this question, the authors use a combination of in vivo and ex vivo experiments to demonstrate the critical role of hemodynamic forces on Notch1 activation and provide strong evidence for a novel mechanotransduction pathway involving PKC and mTORC2.

Strengths:

(1) Novel insights into the role of PKC and mTOR were obtained using a combination of mutant studies and pharmacological studies.

(2) Novel insights on the role of mechanical forces on caveolin-1 localisation.

(3) Mechanical forces were manipulated using the class III antiarrhythmic drug dofetilide, which transiently blocks heartbeat. Care was taken to minimise the confounding effects of hypoxia.

Weaknesses:

The authors suggest that shear stress activates the mTORC2-PKC-Notch signalling pathway by altering the membrane lipid microstructure. Although this is a fascinating hypothesis, more evidence will be needed to prove this. In particular, it is not clear how the general addition of cholesterol in dofetilide-treated hearts would result in a rescue of regionalized membrane distribution within the AVC and in high-shear stress areas.

---

## [Referee Report · Reviewer #3 (Public review)]

Summary:

The overall goal of this manuscript is to understand how Notch signaling is activated in specific regions of the endocardium, including the OFT and AVC, that undergo EMT to form the endocardial cushions. Using dofetilide to transiently block circulation in E9.5 mice, the authors show that Notch receptor cleavage still occurs in the valve-forming regions due to mechanical sheer stress as Notch ligand expression and oxygen levels are unaffected. The authors go on to show that changes in lipid membrane structure activate mTOR signaling, which causes phosphorylation of PKC and Notch receptor cleavage. The data are largely convincing and support their hypothesis. The conclusions are also novel and significantly add to the field of endocardial cushion biology.

The strengths of the manuscript include the dual pharmacological and genetic approaches to block blood flow in the mouse, the inclusion of many controls including those for hypoxia, the quality of the imaging, and the clarity of the text. In the revision, the authors put forth a good faith effort to address experimentally or textually the concerns of the reviewers. Most weaknesses that were identified in the first submission were addressed and the main claims are convincing. In general, the authors achieved their aims and the results support their conclusions.

---

## [Author Response]

The following is the authors’ response to the original reviews.

**Public Review:**
The overall goal of this manuscript is to understand how Notch signaling is activated in specific regions of the endocardium, including the OFT and AVC, that undergo EMT to form the endocardial cushions. Using dofetilide to transiently block circulation in E9.5 mice, the authors show that Notch receptor cleavage still occurs in the valve-forming regions due to mechanical sheer stress as Notch ligand expression and oxygen levels are unaffected. The authors go on to show that changes in lipid membrane structure activate mTOR signaling, which causes phosphorylation of PKC and Notch receptor cleavage.The strengths of the manuscript include the dual pharmacological and genetic approaches to block blood flow in the mouse, the inclusion of many controls including those for hypoxia, the quality of the imaging, and the clarity of the text. However, several weaknesses were noted surrounding the main claims where the supporting data are incomplete.PKC - Notch1 activation:(1) Does deletion of Prkce and Prkch affect blood flow, and if so, might that be suppressing Notch1 activation indirectly?

To address this concern, we performed echocardiography of *Prkce+/-*;*Prkch+/-*, *Prkce-/-*;*Prkch+/-*, and *Prkce+/-*;*Prkch-/-* mouse hearts (Figure 3-supplement figure 2D), showing no significant effect in heartbeat and blood flow. (Line 308)

(2) It would be helpful to visualize the expression of prkce and prkch by in situ hybridization in E9.5 embryos.

We now added immunofluorescence staining results for both PKCE and PKCH as shown in Figure 3-supplement figure 2B. In E9.5 embryonic heart, PKCH is mainly expressed in the endocardium overlying AV canal and the base of trabeculae, overlapping with the expression pattern of NICD and pPKC^Ser660^. PKCE is expressed in both endocardium and myocardium. In the endocardium, PKCE is mainly expressed in the endocardium overlying AV canal (Line312-314)

(2) PMA experiments: Line 223-224: A major concern is related to the conclusion that "blood flow activates Notch in the cushion endocardium via the mTORC2-PKC signaling pathway". To make that claim, the authors show that a pharmacological activation with a potent PKC activator, PMA, rescues NICD levels in the AVC in dofetilide-treated embryos. This claim would also need proof that a lack of blood flow alters the activity of mTORC2 to phosphorylate the targets of PKC phosphorylation. Also, this observation does not explain the link between PKC activity and Notch activation.

Both AKT Ser473 and PKC Ser660 are well characterized phosphorylation sites regulated by mTORC2 (Baffi TR et. al, mTORC2 controls the activity of PKC and Akt by phosphorylating a conserved TOR interaction motif. Sci Signal. 2021;14.). pAKT^Ser473^ is widely used as an indicator of mTORC2 activity. Therefore, the reduced staining intensity of pAKT^Ser473^ and pPKC^Ser660^ observed in the dofetilide treated embryos should reflect the reduced activity of their common upstream activator mTORC2. This information is provided in Line 317-321.

As PMA is a well-characterized specific activator of PKC, we believe the rescue of NICD by PMA could explain the link between PKC activity and Notch activation.

(3) In addition, the authors hypothesise that shear stress lies upstream of PKC and Notch activation, and that because shear stress is highest at the valve-forming regions, PKC and Notch activity is localised to the valve-forming regions. Since PMA treatment affects the entire endocardium which expresses Notch1, NICD should be seen in areas outside of the AVC in the PMA+dofetilide condition. Please clarify.

As shown in Figure 3C and Figure 3-supplement figure 2B, pPKC, PKCH and PKCE expression are all confined in the AVC region. This explains PMA activates NICD specifically in the valve-forming region. This information is added in Line 312-314.

Lipid Membrane:(1) It is not clear how the authors think that the addition of cholesterol changes the lipid membrane structure or alters Cav-1 distribution. Can this be addressed? Does adding cholesterol make the membrane more stiff? Does increased stiffness result from higher shear stress?

We do not know how exactly addition of cholesterol alters membrane structure and influence mTORC2-PKC-Notch signaling. As cholesterol is an important component of lipid raft and caveolae, it is possible that enrichment of cholesterol might alter the membrane structure to make the lipid raft structure less dependent on sheer stress. This hypothesis need to be tested in further in vitro studies. This information is added to Line 433-436.

(2) The loss of blood flow apparently affects Cav1 membrane localization and causes a redistribution from the luminal compartment to lateral cell adhesion sites. Cholesterol treatment of dofetilide-treated hearts (lacking blood flow) rescued Cav1 localization to luminal membrane microdomains and rescued NICD expression. It remains unclear how the general addition of cholesterol would result in a rescue of regionalized membrane distribution within the AVC and in high-shear stress areas.

We do not know the exact mechanism. As replied in the previous question, future cell-based work is needed to address these important questions. (Line 433-436)

(3) The authors do not show the entire heart in that rescue treatment condition (cholesterol in dofetilide-treated hearts). Also, there is no quantification of that rescue in Figure 4B. Currently, only overview images of the heart are shown but high-resolution images on a subcellular scale (such as electron microscopy) are needed to resolve and show membrane microdomains of caveolae with Cav1 distribution. This is important because Cav-1could have functions independent of caveolae.

In Figure 4C, most panels display the large part of the heart including AVC, atrium and ventricle. The images in the third column appear to be more restricted to AVC. We have now replaced these images to reveal AVC and part of the atrium and ventricle.

The quantification has also been provided in Figure 4C. We also added a new panel of scanning EM of AVC endocardium, showing numerous membrane invaginations on the luminal surface of the endocardial cells. The size of the invaginations ranges from 50 to 100 nm, consistent with the reported size of caveolae. Dofetilide significantly reduced the number of membrane invaginations, which recovered after restore of blood flow at 5 hours post dofetilide treatment. The reduction of membrane invaginations could also be rescued by ex vivo cholesterol treatment. This information is added to Line 342-349.

Figure Legends, missing data, and clarity:(1) The number of embryos used in each experiment is not clear in the text or figure legends. In general, figure legends are incomplete (for instance in Figure 1).

Thanks for reminding. we have now added numbers of embryos in the figure legends.

(2) Line 204: The authors refer to unpublished endocardial RNAseq data from E9.5 embryos. These data must be provided with this manuscript if it is referred to in any way in the text.

The RNAseq data of PKC isoforms is now provided in Figure3-Figure supplement 2A, Line 301-302.

(3) Figure 1 shows Dll4 transcript levels, which do not necessarily correlate with protein levels. It would be important to show quantifications of these patterns as Notch/Dll4 levels are cycling and may vary with time and between different hearts.

The Dll4 immuno-staining in Figure 1B,C is indeed Dll4 protein, not transcript. The quantification is added in Figure 1—Figure supplement 1C. Line 215.

(4) Line 212-214: The authors describe cardiac cushion defects due to the loss of blood flow and refer to some quantifications that are not completely shown in Figure 3. For instance, quantifications for cushion cellularity and cardiac defects at three hours (after the start of treatment?) are missing.

The formation of the defects is a developmental process and time dependent. To address this concern, we quantified the cushion cellularity at 5 hours post dofetilide treatment and showed that cell density significantly decreased in the dofetilide treated embryos, albeit less pronounced than the difference at E10.5. (Line 256-257)

(5) Related to Figure 5. The work would be strengthened by quantification of the effects of dofetilide and verapamil on heartbeat at the doses applied. Is the verapamil dosage used here similar to the dose used in the clinic?

We are grateful to this suggestion. The effect of dofetilide on heartbeat has already been shown in Figure 2A. We have now additionally measured the heartbeat rate of verapamil treated embryos, and provided the results in Figure 5E. For verapamil injection in mice, a single i.p. dose of 15 mg/kg was used, which is equivalent to 53 mg/m^2^ body surface. Verapamil is used in the clinic at dosage ranging from 200 to 480 mg/day, equivalent to 3.33 - 8 mg/kg or 117 - 282 mg/m^2^ body surface. Therefore, the dosage used in the mouse is not excessively high compared to the clinic uses. (Line 361-365)

Overstated Claims:(1) The authors claim that the lipid microstructure/mTORC2/PKC/Notch pathway is responsive to shear stress, rather than other mechanical forces or myocardial function. Their conclusions seem to be extrapolated from various in vitro studies using non-endocardial cells. To solidify this claim, the authors would need additional biomechanical data, which could be obtained via theoretical modelling or using mouse heart valve explants. This issue could also be addressed by the authors simply softening their conclusions.

We aggrege with the reviewer’s comment. We have now revised the statement as “Our data support a model that membrane lipid microdomain acts as a shear stress sensor and transduces the mechanical cue to activate intracellular mTORC2-PKC-Notch signaling pathway in the developing endocardium. (line 416-418) It is noteworthy that the methodology used to alter blood flow in this study inevitably affects myocardial contraction. Additional work to uncouple sheer stress with other changes of mechanical properties of the myocardium with the aid of theoretical modelling or using mouse heart valve explants is needed to fully characterize the effect of sheer stress on mouse endocardial development.” (Line 436-440)

(2) Line 263-264: In the discussion, the authors conclude that "Strong fluid shear stress in the AVC and OFT promotes the formation of caveolae on the luminal surface of the endocardial cells, which enhances PKCε phosphorylation by mTORC2." This link was shown rather indirectly, rather than by direct evidence, and therefore the conclusion should be softened. For example, the authors could state that their data are consistent with this model.

We have revised the statement as “Strong fluid shear stress in the AVC and OFT enhances PKC phosphorylation by mTORC2 possibly by maintaining a particular membrane microstructure.” (Line 372-374)

(3) In the Discussion, it says: "Mammalian embryonic endocardium undergoes extensive EMT to form valve primordia while zebrafish valves are primarily the product of endocardial infolding (Duchemin et al., 2019)." In the paper cited, Duchemin and colleagues described the formation of the zebrafish outflow tract valve. The zebrafish atrioventricular valve primordia is formed via partial EMT through Dll-Notch signaling (Paolini et al. Cell Reports 2021) and the collective cell migration of endocardial cells into the cardiac jelly. Then, a small subset of cells that have migrated into the cardiac jelly give rise to the valve interstitial cells, while the remainder undergo mesenchymal-to-endothelial transition and become endothelial cells that line the sinus of the atrioventricular valve (Chow et al., doi: 10.1371/journal.pbio.3001505). The authors should modify this part of the Discussion and cite the relevant zebrafish literature.

Thanks for valuable comments. We have now revised the statement as “Mammalian embryonic endocardium undergoes extensive EMT to form valve primordia while zebrafish atrioventricular valve primordia is formed via partial EMT and the collective cell migration of endocardial cells into the cardiac jelly followed by tissue sheet delamination.” with relevant references added. (Line 411-414)

**Recommendations to the Authors:**
(1) One issue that the authors could address is the organization of figures. There are several cases where positive data that are central to the conclusions are placed in the supplement and should be moved to the main figures. Places where this occurred are listed below:- The Tie2 conditional deletion of Dll4 showing retention of NICD in the OFT and AVC regions is highly supportive of the model. The authors should consider moving these data to main Figure 1.

Thanks for the suggestion. We have reorganized the figure as requested.

- The ligand expression data in Figure 2- Supplement Figure 1 A is VERY important to the conclusions drawn from the dofetilide treatment. The authors should move these data to main Figure 2.

The ligand expression data in Figure 2- Supplement Figure 1A are now moved to Figure 2B.

- In Figure 3A - the area in the field of view should be stated in the Figure (is it the AVC?) Figure 3 - Supplement 1 proximal OFT data should be moved to main Figure 3 as it is central to the conclusions. Negative DA data can be left in the supplement. Again, for Figure 3 - Supplement 1 Stauroporine treatment data should be moved to the main figure as it is positive data that are central to the conclusions.

Thanks for the suggestion. We have reorganized the figure as requested.

(2) Antibody used for Twist1 detection is not listed in the resource table.

Twist1 is purchased from abcam, the detailed information is now available in the resource table.

(3) Missing arrowhead in Figure 4A, last row.

Sorry for the negligence. Arrowhead is now added.

(4) Line 286. "OFT" pasted on the word "endothelium".

“OFT” is now removed.

(5) Related to Figure 2C. The fast response of NICD to flow cessation was used as an argument to support post-translational modification. It is not clear why Sox9 and Twist1 expression also responds so quickly.

Sox9 and Twist1 expression does seem to respond very quickly. Whether there exists additional regulatory pathways such as Wnt, Vegf signaling that also respond to sheer stress needs to be investigated in the future.

(6) Line 200: The sentence should end with a period.

Sorry for the oversight. It is now corrected.

(7) Lines 34 to 35: the authors phrase that Notch is "allowed" to be specifically activated in the AVC and outflow tract by shear stress.

We have rephrased the statement with “enabling Notch to be specifically activated in AVC and OFT by regional increased shear stress.” Line 27

(8) Lines 96-100: At the end of the introduction, the text is copied from the abstract. New text should be written or summarized in a different way.

The last sentence of introduction is now changed to “The results uncovered a new mechanism whereby mechanical force serves as a primary cue for endocardial patterning in mammalian embryonic heart.” (Line 93-95)

(9) Line 125: The term "agreed with the Dll4 transcript.."should be replaced with a better term like "overlapped" or "was identical with".

The word “agreed” is now “overlapped”. (Line 219)

(10) Line 291: "Thus, through these sophisticated mechanisms, the developing mouse hearts may achieve three purposes:"- The English should be adjusted here since it sounds like hearts are aiming to achieve a purpose, which is unlikely what was meant by the authors.

This sentence is rephrased to “Thus, in the developing mouse hearts: (1) VEGF signaling is reduced to permit endocardial EMT; (2) Dll4 expression is reduced to prevent widespread endocardial Notch activation and make endocardium sensitive to flow; (3) a proper cushion size and shape is maintained by limiting the flanking endocardium to undergo EMT despite physically close to the field of BMP2 derived from of AVC myocardium (Figure 6).” (Line 402-406)